# Tensile Properties and Fracture Mechanism of Thermal Spraying Polyurea

**DOI:** 10.3390/polym15010041

**Published:** 2022-12-22

**Authors:** Haotian Zhang, Yongyuan You, Yongsheng Jia, Jianian Hu, Peibo Li, Quanmin Xie

**Affiliations:** 1State Key Laboratory of Precision Blasting, Jianghan University, Wuhan 430056, China; 2Hubei Key Laboratory of Blasting Engineering of Jianghan University, Wuhan 430056, China; 3State Key Laboratory of Advanced Technology for Materials Synthesis and Processing, Wuhan University of Technology, Wuhan 430070, China

**Keywords:** polyurea, fracture performance, cracks, pores, spraying pressure, spraying temperature

## Abstract

In this study, polyurea was experimentally tested under various spraying temperatures and pressures. The number of holes and the pore size produced after the tensile fracture of the polyurea were counted to illustrate the effect of the various spraying temperatures and pressures on the performance of the polyurea. The tensile characteristics of polyurea were greatly influenced by the spraying temperatures and pressures, according to the experimental findings and statistical analysis. The polyurea tensile performance was best when the spraying pressure was 17.25 MPa with a spraying temperature of 70 °C. The fracture mechanism was illustrated by the silver streaking phenomenon generated during the tensile stretching process. The fracture energy was absorbed by the fracture holes and pores during silver streaking, thus creating the huge gap in tensile properties.

## 1. Introduction

Elastomeric materials with exceptional qualities have been created in the last several decades to suit the expanding demand for use in industry and in national defense [1,2]. Due to its excellent physical and chemical properties, high thermal stability, the material’s lack of weight loss until 225 °C, lightweight density (1.02 g/cm^3^), corrosion resistance, abrasion resistance, nontoxicity, low price, and other excellent characteristics of excellent stability in complex operating environments, polyurea elastomer material has gradually gained attention in recent years [3,4,5,6,7,8]. These excellent characteristics have a significant impact in the defense sector and the military industry [9].

Fragiadakis et al. [10] investigated the correlation between the chemical ratio between the soft and hard segments in polyurea and the mechanical characteristics of polyurea elastomer molecules. The limited deformation stress–strain behavior of a polyurea elastomer under uniaxial compression is temperature-dependent and highly nonlinear, according to Chen et al. [11] investigated the mechanical properties of polyurea elastomer materials at various temperatures. The researchers separated the Hopkinson pressure bar test to obtain the stress–strain behavior of polyurea elastomer with a strain rate correlation under uniaxial compression load and restricted pressure. Additionally, considerable research was done into how sensitive polyurea elastomers are to strain rates, both at low and high rates. To improve the tensile properties of polyurea elastomer materials, the spraying temperature and pressure must be controlled before the materials are cured [6]. The mechanical properties of polyurea materials have been extensively researched after spraying, but research on controlling the spraying temperature and pressure during the initial spraying process has not been reported.

To broaden the scope of application, numerous academics have recently examined the mechanical characteristics of polyurea materials [12]. Although polyurea performs well, its application is restricted. As a result, the polyurea structure is modified using a variety of surface modification techniques to achieve improved performance [13,14]. However, the initial spraying temperature and pressure have received little consideration. Due to the impact mixing technology of polyurea spraying, two liquids (A component and B component) with extremely high reactivity bump into one another under high temperatures and high pressures [15]. The two liquids are then dispersed and evenly mixed in a very small mixing chamber with turbulence, and then they are atomized and mixed again after being extruded from the gun under high pressure [16,17]. The hydrogen bonding content between the soft and hard segments of the polyurea elastomer differs due to the high sensitivity of the polyurea polymer to pressure and temperature. This has a significant impact on the microstructure and mechanical properties. As the temperature and pressure increase, the polymer chain movement also increases [18,19,20,21]. As a result, during the spraying process, we can adjust the spraying pressure and spraying temperature to find the ideal conditions for optimizing the mechanical properties of polyurea elastomers, which is unquestionably necessary given the level of tensile, compression, blast, and impact resistance of the material.

In this paper, the quasi-static tensile characteristics of polyurea at different spraying pressures and temperatures were studied. The microstructure of polyurea dolphin was further refined by scanning electron microscope (SEM), and the relationship between the number of tensile fracture holes and pore diameter of the material and the tensile performance was revealed. The purpose of this study was to comprehend how the tensile characteristics of polyurea are affected by various pressures and temperatures. This is required for the efficient and safe use of such polyurea materials in applications with various operating conditions.

## 2. Materials and Methods

### 2.1. Materials Synthesis

In the present study, the spraying process of polyurea (Qtech-413, Qingdao Shamu Advanced Material Co., Ltd., Qingdao, China) tensile specimen was completed in the State Key Laboratory of Precision Blasting Engineering, Jianghan University. A substance known as polyurea elastomer was created by stepwise addition polymerizing isocyanate and ammonia molecules. The B component was a two-component elastomer material made of amino-terminated polyether, hydroxyl-terminated polyether, amine chain extenders, and auxiliary agents, while the polyurea A component was a semiprepolymer containing terminal NCO groups (Qingdao Shamu New Material Co., Ltd., Qingdao, China).

To create polyurea tensile test pieces, A and B components were combined and put into a spray gun, then sprayed on the drawing die at various temperatures and pressures. The synthesis process and stretching of the polyurea specimen is shown in Figure 1. According to research by Tuerp D et al. [22,23], AB components have excellent physical and chemical properties when combined at 60–90 °C, so we prepared four tensile specimens with four different spraying pressures (12.42 MPa, 15.53 MPa, 17.25 MPa, 18.98 MPa) at 70 °C ,and four different spraying temperatures (55 °C, 63 °C, 70 °C, 80 °C) at a spraying pressure of 17.25 MPa. The polyurea sheets were prepared by spray-on procedure and the cast sheets were allowed to cure at room temperature. The test specimens were die-cut from the cast sheets to the dimensions shown in Figure 1 (with an allowance of ±0.2 mm). The specimens were then ground to smoothen their surfaces and to a thickness of 3 mm. The three-dimensional size and appearance of the polyurea samples are shown in Figure 2a,b. 

### 2.2. Tensile Test Systems

The polyurea material was removed after curing at room temperature for at least two weeks before the test and all tests were conducted at room temperature (approximately 25 °C). This prevented the relative slippage between polyurea and the upper/lower grid in the tensile experiment, resulting in a large experimental error, so the two ends of the material were subtracted by 50 mm before the experiment. The experimental procedures were adapted from the guidelines provided in ASTM: D412 and were customized based on the setup and capacity of the testing systems [3]. A universal tensile machine (type 68TM-30, Instron, Boston, USA) was applied for the tensile test under constant displacement speed of 30 mm/min, and the load cell was 2580-10kn type of Instron. The Instron tensile testing system shows in Figure 2c. Sensors on the upper and lower grips of the high-speed Instron test system were used to record the time histories of displacement and force. The tensile stress–strain characteristics of polyurea were resolved to utilize these datasets.

### 2.3. Scanning Electron Microscopy (SEM) and ImageJ

Before testing, samples were sputter-coated with a gold ion beam for 100 s tests to enhance electrical conductibility. In this study, the samples were taken using scanning electron microscopy (SEM, Phenom Pure, Shanghai, China) to capture surface topography images.

The polyurea material was stretched using a universal stretching machine to obtain the tensile fracture morphology image of the specimen after the tensile test. The size and quantity of holes produced after the polyurea material was stretched were measured using ImageJ software (ImageJ is a public image processing software based on java) in the tensile fracture morphology image. The statistics are displayed in Figure 3.

## 3. Results and Discussion

### 3.1. Static Tensile Results

Figure 4 shows the tensile stress–strain curves for the polyurea material from the quasi-static tensile test. The original cross-sectional area and specimen length are utilized to calculate the stress–strain curve. The tensile stress–strain curve of polyurea is nonlinear, with high strength and high elongation, which resembles that of a typical elastic–plastic material. The material initial displayed a linear region and then experiences a short yield period. At the point when further loads are added, the material then experiences significant elongation (plastic deformation), and the applied stress gradually rises. The material undergone a strain-hardening procedure as the curve ascends. The polyurea shows an almost straight bend around here. Ichinomiya et al. [24] indicated that throughout the yielding process, the material’s microstructure changes, and the substance yields as the tiny pores leak out. Figure 4 shows that the stress–strain curve will have a near plateau segment, a concave point, and then rise to a peak. Due to defects in polyurea materials during tensile and the inevitable bonding sliding between the end of the sample and the test machine fixture during the tensile process.

In Figure 4a, the material’s lowest tensile stress was 2.2 MPa when it produced the greatest tensile strength of 3.83 MPa. However, when pressure grew, the yield stress fluctuated up and down, which also dramatically altered the yield strength magnitude. The pressure of the polyurea material had a sizable impact on the yield stress of the material during the spraying operation. When the spraying pressure was between 12.42 MPa and 17.25 MPa, the best impact of material yielding was seen, while the other two spraying pressures caused varying degrees of decline. The spray pressure will no longer substantially impact the material’s yield strength after it reaches 18.98 MPa. The highest tensile stress for material yield, shown in Figure 4b, is 3.75 MPa, whereas the minimum tensile stress is 2.26 MPa. The temperature of the material during the spraying process has a considerable impact on the yield stress of the polyurea material. The tension when the material gives increases with the spraying temperature. The yield strength of the polyurea material was significantly increased when the spraying temperature was raised from 55 °C to 80 °C, and the polyurea characteristics varied with the amplitude change as the temperature rose. As seen in Figure 4, the material yielded about equal tensile stress and strain, practically instantaneously, and a change in elongation that was commensurate with the tensile stress. The maximum and lowest values of the material’s yield stress are comparable under various pressure and temperature circumstances, which is consistent with the polyurea elastomer’s yielding properties. Changes in spraying pressure and temperature affect the molecular chain structure (especially hydrogen bonds) inside the material to varying degrees, resulting in different tensile strength of the material. After a short yield, the material will undergo a long period of plastic deformation. During this period, the slope of the tensile stress–strain curve is roughly the same, but with the increase of pressure and temperature, there are obvious fluctuations and deviations in elongation and yield stress.

According to the stress–strain curve of the tensile experiment, the obvious trend cannot be seen in Figure 4, so we have listed and compared the peak tensile results of polyurea materials in Figure 5. The changes in spraying pressure and spraying temperature affect the change of the tensile strength of the material. It can be seen in Figure 4b that the stress–strain curve of the material at 55 has a sharp rise stage due to secondary loading, so the peak experimental results before secondary loading are selected to compare the tensile strength of the material. In practice, polyurea can be sprayed according to different use environments and different working conditions.

Figure 5 illustrated the results of the polyurea material’s tensile stress and strain. Figure 5a demonstrates that the tensile strain of the polyurea material gradually increases with increasing pressure. The tensile effect of the material reaches its maximum when it reaches 17.25 MPa, and the tensile strain of the polyurea material is 648.79%, which confirms the previously described elongation correlation with the spraying pressure when the material exhibits the best performance. When the spray pressure was maintained at 17.25 MPa, the polyurea material had the greatest tensile quality in terms of elongation and tensile strength. The tensile strain of the polyurea material changed with increasing spray pressure. The polyurea material’s maximum tensile strain at 70 °C, as shown in Figure 5b. The tensile strain of the polyurea material is at least 458.79% when the spraying temperature is 55 °C. Only when the temperature of the spray was kept between 55 °C and 70 °C does the elongation of the polyurea material reach its maximum. Therefore, when the spraying temperature is 70 °C, the material was the best tensile characteristics and the best tensile stress–strain match. The tensile stress–strain results in Figure 5a,b are based on HSD Tukey analysis. Based on the average value *p* = 0.046 < 0.05 in Figure 5a shows that the polyurea material changes the tensile strength of the material under the same spraying temperature and different spray pressure. Figure 5b based on the average value *p* = 0.053 > 0.05 shows that the polyurea material changes the spray temperature under different spraying pressure.

### 3.2. Fracture Mechanism

Polymer microstructure and macroscopic fracture behavior are strongly connected. The molecular chains are aligned in the tensile direction when the material is deformed by external tensile pressures, and when the material yields, the conformation of the molecular chains has already experienced a significant change [25]. Following the yielding deformation process, certain chemical bonds in the molecular chains entangle and finally break under stress, leading to the macroscopic fracture of the substance [26]. According to Lin et al. [27], surface stress is the primary cause of polymer deformation. When the surface stress at the crack interface is less than the yield stress, surface stress will occur within or on the material’s surface, causing the polymer to fracture and causing a silver pattern to appear inside or on the material’s surface. The silver grain deformation method will be applied to the polymer. Silver rippling is an early stage of crack creation before material damage occurs, occurring before the macroscopic fracture of polymeric materials. Silver rippling is typically seen at the start of microscopic damage to the material since it makes the surrounding undamaged region more vulnerable to rupture when the material is exposed to tensile pressures [28,29,30]. The development of fractures and holes in polyurea is briefly depicted in Figure 6.

### 3.3. Fracture Results under Different Spray Pressures and Temperatures

Figure 7 depicts the polyurea material’s surface before stretching, which is smooth and dense. The smoother the surface is due to an increase in spraying pressure, and there are no holes or fractures, indicating that there was no gas intervention on the material’s surface. The material exhibits many fractures upon stretching and fracture, has a rough surface, and the SEM picture displays hill laminae due to the non-flat fracture surface. In Figure 7, the fracture surface of the polyurea material is where the broken piece first appears after being stretched to create a circuit.

Figure 7d shows that the polyurea material’s surface is smooth and devoid of any pleated lumpy morphology, which is entirely associated with the rise in pressure. Figure 7e,h show that the polyurea material exhibits a significant number of fine surface cracks at the fracture surface after stretching. More than 90% of these cracks bypass the pores, and Figure 7h shows that the material’s toughness is evident despite the fact that few characteristic performances are visible through the pores. The polyurea material’s surface is rough in Figure 7b, and Figure 7f shows a significant amount of overall ridge laminar folding, which is the same shape as the rock-like brittle material when fracture occurs. The cracks at the tensile fracture are large, and there are cracks on the surface of the fracture, and the material’s brittleness is evident under this pressure condition. The bumps are greatly lessened in Figure 7c, and certain places that experienced tensile fracture show laminar folding as in Figure 7g. The material itself has a very high elongation due to the increased toughness, the consistent distribution of hole sizes throughout the material, and the difficulty in preventing perforation when fracture does ultimately occur. Maintaining elastic characteristics under pressure is advantageous for the substance.

As can be seen in Figure 7e–h that as spraying pressure increases, all the materials exhibit laminar characteristics of brittle materials resembling rock. The brittleness of the materials is strongly reflected at the moment of fracture, and the tensile stress is completely released in an instant. With the increase of spraying pressure, the reduction of crack perforation through Figure 7e,h will increase the tensile strength of the materials by 16.5%. This is consistent with the results already discussed.

The material breaks down as a result of the tensile fracture process, in which the majority of the energy is absorbed by the holes and cracks. When the material surface is not smooth before spraying, it will develop various degrees of lumps, folds, and rough surface morphology throughout the curing process. When the spraying temperature changes, the material surface produces various morphological changes, which are most noticeable in Figure 8c. As the spraying temperature increases, the surface of the material becomes smoother. The material is more brittle at the tensile fracture the more pronounced the ridge laminae are, while the spraying temperature steadily rises. The material exhibits comparable fracture morphology in Figure 8e,f, where fractures typically manifest around holes and seldom penetrate them. Despite this, the material nevertheless exhibits some toughness. Figure 8g,h illustrate a fracture morphology with a wide laminar pattern in the form of cracks that mostly develop via the perforations. These fractures will partially resemble loops from Figure 8, although there is considerable variation in them. As a result, with the increase of spraying temperature, different forms of changes will occur at the cross section of the material fracture, and the critical point for optimal tensile performance of the material is between 63 °C and 70 °C.

### 3.4. Number of Fracture Pores and Distribution 

Figure 9 shows that the number of tensile fracture holes in the material is different due to the difference between spraying pressure and spraying temperature, which will directly affect the tensile performance of the material. The holes and cracks generated by the material during the tensile fracture absorb most of the tensile force. After reaching the critical tensile strength of the material, the material will eventually break. Therefore, the tensile resistance of the material under different spray conditions can be further proved according to the number of fracture holes in the material.

According to Figure 9a, the material’s number of holes reaches its maximum at a spraying pressure of 17.25 MPa. This indicates that the material’s holes are absorbing the most energy at this point, it indicates that the material’s tensile performance is at its best. The performance of polyurea will significantly rise along with the number of tensile holes, which has increased by 50.5% when the spraying pressure is increased. The optimum performance of polyurea material is attained when the spraying pressure is kept at roughly 17.25 MPa; however, if the spraying pressure is increased further, the material’s performance degrades. In Figure 9b, the number of holes created by the tensile fracture of polyurea material varies with the increase of spraying temperature. When the spraying temperature rises from 63 °C to 70 °C, the number of holes dramatically increases by 48.4%. Additionally, it demonstrates that polyurea material performs best at a temperature of 70 °C for spraying. The performance of the material will improve once the optimal spraying pressure and temperature are reached, and the performance of the polyurea material will bring a decreasing trend after the optimal spraying pressure and temperature, according to Figure 9. When the spraying pressure is 17.25 MPa and the temperature is 70 °C, the polyurea material performs at its best, as shown in Figure 5 and Figure 9.

The interior of the polyurea material is a porous structure. After the material is stretched, the original hole absorbs the energy generated during stretching. The pore size will be further increased, and silver sprinting occurs inside the material at the same time to produce new holes and cracks. According to the different combinations of spraying pressures and spraying temperatures, the number of fracture holes and pore sizes of the material will be different, which will affect the different tensile performances of the material. Figure 10 shows the distribution of pore size at the tensile fracture of the material. According to Figure 10, the fracture hole is mainly distributed between 15 and 35 µm, and this part of the hole absorbs more than half of the fracture energy during the tensile process. This part of the hole absorbs excellent fracture energy. In Figure 10, when the aperture is greater than 60 µm, the number of holes will decrease sharply. This part of the hole (about 3% of the total number of holes) has considerable absorption potential. The large aperture absorbs more energy, but the overall number is very small. In Figure 10a, when the spraying temperature is 70 °C and the spraying pressure is 17.25 MPa, the aperture distribution is more concentrated, so that the tensile fracture energy is absorbed more uniform so that the elongation of the material is higher, and the tensile resistance of the material is increased. Figure 10b depicts the distribution of tensile fracture apertures at different spraying temperatures of the same spraying pressure. At the spraying temperature of 70 °C, the aperture is more concentrated, and there are no holes greater than 100 µm. The hole distribution and pore size are excellent, which is the phase observed when the spraying pressure in Figure 10a is 17.25 MPa. Combined with the above discussion, it was concluded that the tensile performance of the material for lifting the spraying pressure and spray temperature was greatly improved. When the spraying pressure is 17.25 MPa and the spraying temperature is 70 °C, the polyurea material has the best tensile resistance and tensile effect.

We considered a polyurea tensile fracture specimen where the crack terminates in a circular cavity with a diameter d that is very small compared to the crack length within a planar strain elastic framework, which in turn indicates the toughness enhancement of the material by the number and size of the pore size in Figure 11a. In this study, the pore size and porosity effects are taken into account, and the material section is reduced to a 2D model [31]. 

To clarify the relationship between the tensile properties of materials and critical holes and cracks. Thus, the solution ${\underline{U}}^{d}$ to the elastic problem is expressed as the solution ${\underline{U}}^{0}$ to the unperturbed problem plus a correction, the smaller the pore diameter, the smaller the correction.
(1)$${\underline{U}}^{d}\left(x_{1},x_{2}\right)={\underline{U}}^{0}\left(x_{1},x_{2}\right)+small\;correction$$

The leading term ${\underline{U}}^{0}$ is the far field; it is a satisfying approximation of the actual solution ${\underline{U}}^{d}$ except in the vicinity of the perturbation. Under the condition of small parameters d and stretched variables $y_{i}=x_{i}/d$, the elastic solution expands to generate an internal expansion.
(2)$${\underline{U}}^{d}\left(x_{1},x_{2}\right)={\underline{U}}^{d}\left(dy_{1},dy_{2}\right)=F_{0}\left(d\right){\underline{V}}^{0}\left(y_{1},y_{2}\right)+F_{1}\left(d\right){\underline{V}}^{1}\left(y_{1},y_{2}\right)+\ldots with\;\underset{d\to 0}{\lim}\frac{F_{1}\left(d\right)}{F_{0}\left(d\right)}=0$$

Then
(3)$${\underline{U}}^{0}\left(x_{1},x_{2}\right)={\underline{U}}^{0}\left(0,0\right)+k_{1}\sqrt{r}u^{I}+\ldots$$
where r and $u^{I}$ are the polar coordinates with the origin at the crack tip. The mode I stress intensity component of the traditional square root behavior of the opening mode is represented by the coefficient $k_{1}$. The prescribed loading is supposed symmetric in this model. Thus,
(4)$$F_{0}\left(d\right)=1\;\;F_{1}\left(d\right)=k_{1}\sqrt{d}\;\;{\underline{V}}^{0}\left(y_{1},y_{2}\right)={\underline{U}}^{0}\left(0,0\right)\;\;{\underline{V}}^{1}\left(y_{1},y_{2}\right)=\sqrt{\rho }u^{I}\;\mathrm{with}\;\;\rho =r/d\;$$

The best fit was obtained when the pressure was 17.25 MPa and the temperature was 70 °C, indicating the best material toughness and confirming that the material’s elongation was greatest, and its tensile properties were best. This was done after incorporating the tensile result pore size data in Figure 10 and Figure 11. The polyurea tensile fracture demonstrates that round holes predominate in the fracture, as opposed to elliptical holes, which have a passivation effect and diminish the material’s toughness. The porosity V of the material segment was therefore estimated for a more thorough comparison.
(5)$$V=\frac{\pi {\left(\sum_{i=0}^{n}r_{i}\right)}^{2}}{s}$$

Integration: (6)$$F\left(V\right)=\overset{r_{i}}{\underset{0}{\int }}\frac{\pi {\left(\sum_{i=0}^{n}r_{i}\right)}^{2}}{s}dr\;$$
where the SEM image area and pore size are $r_{i}$ and s, respectively. According to the pore size calculation, the maximum porosity is 57.5% at 17.25 MPa and 55.4% at 70 °C. As a result of tip passivation, it is presumable that the main crack’s original position is in the pore. The toughness is concurrently markedly strengthened, and the rate of energy absorption is higher, which results in less material passivation. The wider the pore, the greater the enhancement. The higher the porosity of the material, the higher the toughness. In the discussion above, the material has the best tensile characteristics at 17.25 MPa and 70 °C when the pore size effect is included, and the porosity is contrasted with the trend that was previously mentioned.

## 4. Conclusions

The tensile properties of polyurea materials were examined in this study as well as the formation of holes and cracks during the tensile process. Tensile fracture holes and pores were used to discuss the tensile properties of polyurea at various spraying pressures and temperatures, revealing the fracture mechanism of the polyurea. 

The yield stress of polyurea materials is found to be significantly influenced by the spraying pressure, and the tensile strain is roughly the same when the material yield and the yield occur almost simultaneously. The tensile yield stress achieves its peak at a spray pressure of 12.42 MPa and then constricted to a spray pressure of 17.25 MPa. and the yield performance is best when the spray temperature reaches 80 °C.

The number of fracture holes produced by a tensile fracture and the distribution of pore size are entirely different due to variations in spraying pressure and temperature, which in turn influences the tensile characteristics of polyuria. When the spraying pressure and temperature are 17.25 MPa and 70 °C, respectively, it is evident from the tensile test results, the number of holes, and the results of the pore size distribution that the material has the best tensile properties. The reason is that the pores absorb the most fracture energy during the polyurea stretching process, and the pore size distribution is optimal.

## Figures and Tables

**Figure 1 polymers-15-00041-f001:**
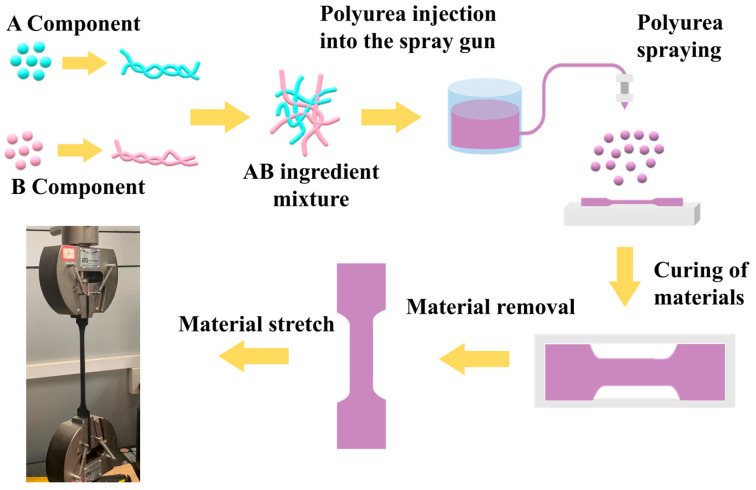
Synthesis process and stretching of the polyurea specimen.

**Figure 2 polymers-15-00041-f002:**
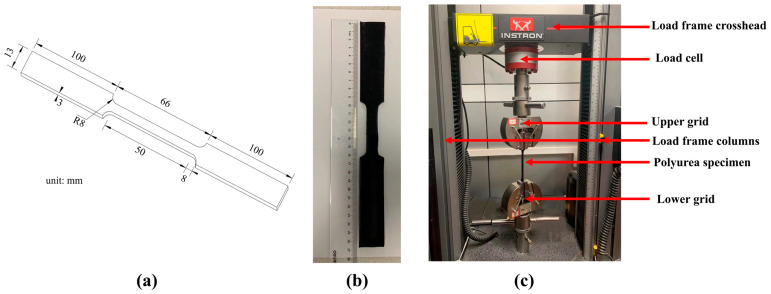
(**a**) tensile specimen outlines sizes; (**b**) appearance of polyurea specimen; and (**c**) the Instron tensile testing system.

**Figure 3 polymers-15-00041-f003:**
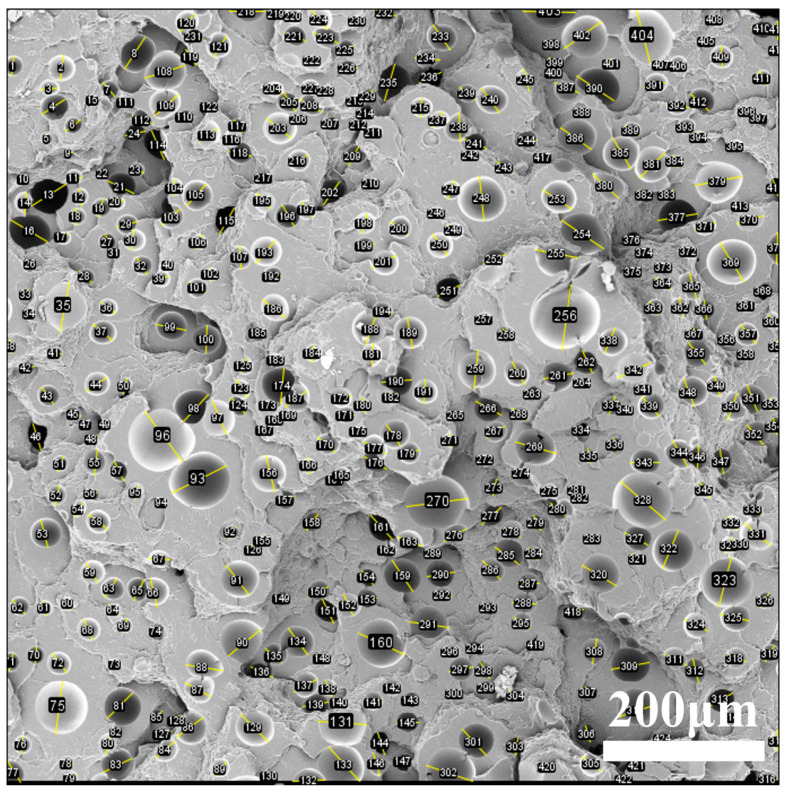
One of the polyurea tensile fracture holes and pore sizes analyzed by ImageJ.

**Figure 4 polymers-15-00041-f004:**
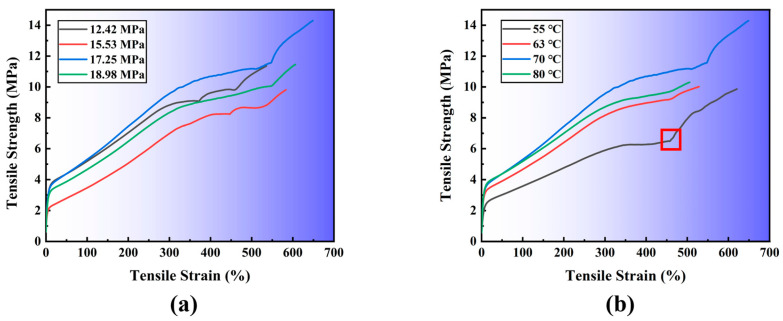
Tensile stress–strain curves of polyurea under different conditions: (**a**) different spraying pressures under the same temperature; and (**b**) different spraying temperatures under the same pressures. (Red Square is due to secondary loading.)

**Figure 5 polymers-15-00041-f005:**
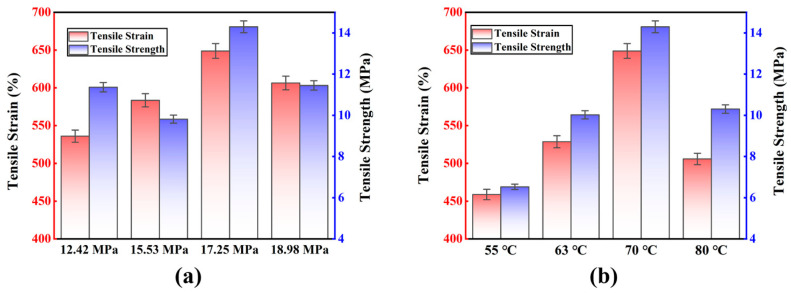
Tensile strain and strength results of polyurea under different conditions: (**a**) Different spraying pressures under the 70 °C; (**b**) Different spraying temperatures under the 17.25 MPa.

**Figure 6 polymers-15-00041-f006:**
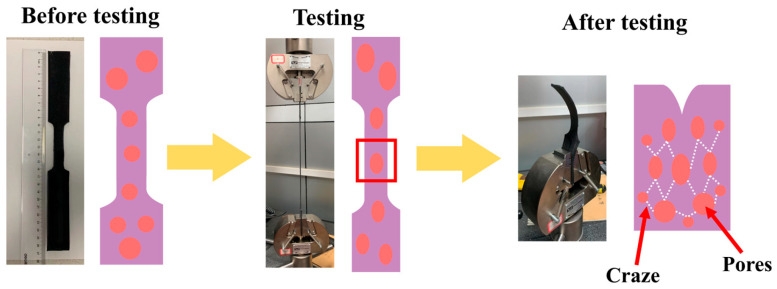
Schematic diagram of polyurea tensile fracture. (Specimen drawing and specimen schematic diagram before testing; Specimen drawing and specimen schematic diagram in testing; Specimen drawing and specimen schematic diagram after testing.)

**Figure 7 polymers-15-00041-f007:**
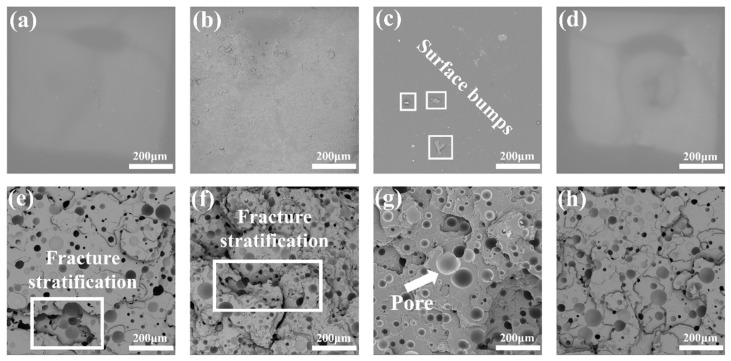
SEM before the surface of polyurea fracture at different pressure under the same temperature: (**a**) 12.42 MPa; (**b**) 15.53 MPa; (**c**) 17.25 MPa; and (**d**) 18.98 MPa. SEM after the cross-section of polyurea fracture t different pressure under the same temperature: (**e**) 12.42 MPa; (**f**) 15.53 MPa; (**g**) 17.25 MPa; and (**h**) 18.98 MPa.

**Figure 8 polymers-15-00041-f008:**
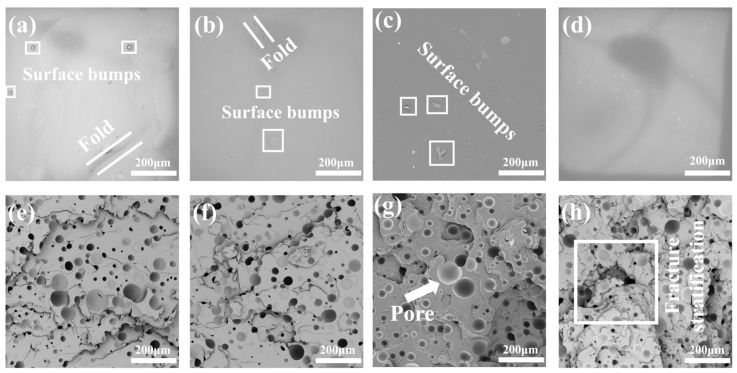
SEM for the front surface of polyurea fracture at different temperature under the same pressures: (**a**) 55 °C; (**b**) 63 °C; (**c**) 70 °C; and (**d**) 80 °C. SEM after the cross-section of polyurea fracture at different temperature under the same pressures: (**e**) 55 °C; (**f**) 63 °C; (**g**) 70 °C; and (**h**) 80 °C.

**Figure 9 polymers-15-00041-f009:**
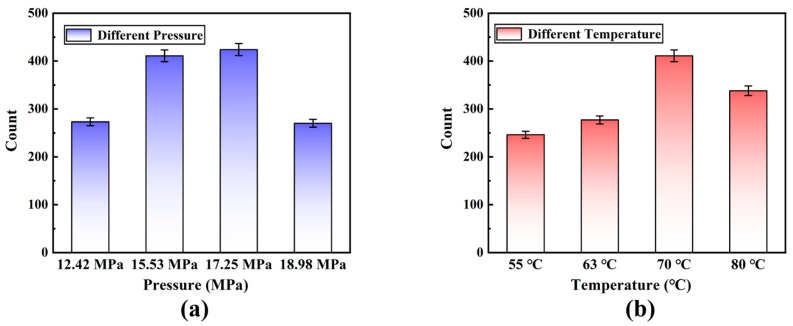
The number of pores counts during tensile fracture of polyurea under different conditions: (**a**) different spraying pressures under the 70 °C; and (**b**) different spraying temperatures under the 17.25 MPa.

**Figure 10 polymers-15-00041-f010:**
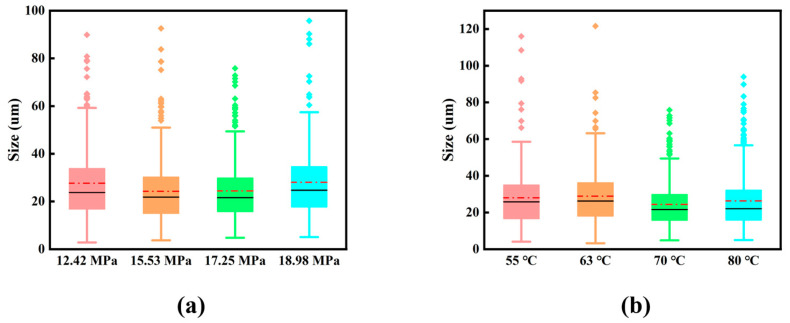
Pore size distribution of polyurea tensile fracture holes: (**a**) different spraying pressures under the 70 °C; and (**b**) different spraying temperatures under the 17.25 MPa.

**Figure 11 polymers-15-00041-f011:**
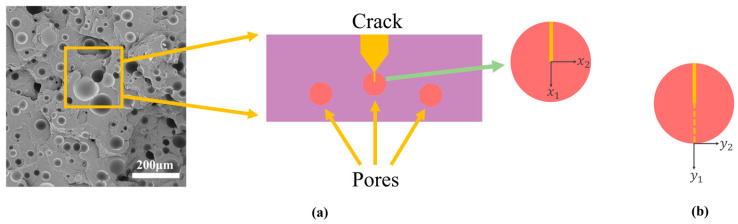
(**a**) The SEM specimen and the pore termination of the primary crack; and (**b**) the stretched single pore.

## Data Availability

The data presented in this study are available on request from the corresponding author.

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
