# Peer review of "Tensile Properties and Fracture Mechanism of Thermal Spraying Polyurea"

_polymers, 2022, doi:10.3390/polym15010041_

Round 1

Reviewer 1 Report

The article presented by H. Zhang on the mechanical( tensile) properties of Polyurea shows interesting results that could be of interest to the Polymers readership. Especially the pore distribution and generation vs the synthesis conditions is an interesting contribution to the topi. However, the following aspects need to be addressed before a positive recommendation.

·         There is much more literature on the mechanical properties of Polyurea to be described in the introduction, and the research mainly cited is outdated. Please address this issue.

·         The last paragraph of the introduction does not properly outline the article's scope.

·         Why Figures 1 and  2 are presented separately? They could easily be figure 1 (a and b). Additionally, I would kindly ask the authors to include a picture of the actual samples, pre and post-testing, and zooms of the failure/fracture sections.

·         Figure 2 is captioned as “Three polyurea tensile fracture holes..” there is a single picture.

·         Figure 5 a and b, would be easily comparable if the y-axis of both images was identical.

·         Please include the appropriate micron symbol in the graphs.

·         Gaussian distribution for Figure 11 b and d, does not seem to be appropriately centred, as it s clearly a skewed distribution (no holes are present below zero). Please reconsider a more suitable distribution. Nevertheless, this could be solved by representing  Figures 10 and 11in a Box-whisker plot. Also, reducing the number of figures in the manuscript and providing higher readability.  

·         One of the main issues I find with the article is that the authors use 4 different pressures (at undefined temperatures) and 4 different temperatures  (at undefined pressure). There is no description of this aspect in the material and methods sections. The study is carried out in a very comprehensive manner with fascinating insights on porous/cavities generations. However, I believe that any reader would ask. –So, should we synthesize our Polyurea by using 17.25 mPa AND 70(deg) celcius?. And the answer, in the manuscript, is “we don’t know, only pressure and only temperature”. It doesn’t mean that the research is invalid, but then, a clear explanation of WHY the author only explored pressure and only temperature would be needed. 

Reviewer 2 Report

The authors have studied the tensile properties and fracture mechanics of polyurea samples, sprayed at varying temperatures and pressures. The current study is an attempt to advance the understanding of mechanical properties of polyurea applied under different application/operational conditions. This work is suited for the materials-oriented audience and for the scope of MDPI polymers. However, before publishing the results, the authors are expected to address several concerns/questions and highlight them in the manuscript. 

After careful reading, I would like to offer my comments as follows.

1.     The authors need to thoroughly check their bibliography, the relevance of their references to the cited text in the main body, and to cite appropriate sentences in the main text. To name a few:

a)     Please add a reference to the following sentence: Elastomeric materials with exceptional qualities have been created in the last several decades to suit the expanding demand for use in industry and national defense

b)    Most of the references [1-6] are pertaining to the mechanical properties of polyurea. Please consider relevant references clearly addressing other chemical and physical properties of polyurea.

c)     Please check reference 21 in the bibliography. It appears that the journal name is missing.

2.     For the sake of reader’s convenience and to tend to a broader audience, please add a discussion about the origin of holes/pores in sprayed polymeric (in this case polyurea) samples during mechanical stretching. 

3.     In the Methods section, please provide elaborate details of the tensile test to facilitate reproducibility à load cell used, stretching cycle steps, ASTM (or similar) standard used, sample type, and similar relevant details.

4.     In Figure 1, the authors need to highlight the ASTM (or similar) standard referenced and the geometry type of the dog-bone schematic.

5.     In Figure 3, the snapshot represents only one polyurea sample. Please use appropriate description for the figure caption and include scale bar for the image.

6.     In Figure 4, are the stress-strain curves representative of a single sample or average of multiple samples? If multiple, please highlight N (# of samples) in the figure caption and include standard deviation in the plots for each curve.

7.     In addition to point 6, there are no clear trends observed in Figure 4a and 4b. How to understand the influence of spraying pressure and temperature on the mechanical properties of polyurea? What are its implications? What do we learn from the application standpoint? What is the reason for the lack of consistent trend in both Figure 4a and 4b?

8.     In Figure 5, are the tensile strain and strength values representative of a single sample or average of multiple samples? If multiple, please highlight N (# of samples) in the figure caption and include standard deviations/errors for each bar plot. Additionally, to enable comparison between treatment groups within Figure 5a and 5b, multiple pairwise comparisons (like HSD Tukey tests) should be performed. The authors are expected to perform statistics to address the above concern and highlight the same in the main manuscript. 

9.     In Section 3.2, are the authors referring to surface energy? Please ensure that correct terminologies are used to convey appropriate scientific explanations/thoughts. Also, it would be convenient for easy comprehension if the authors could provide a schematic representation of silver rippling phenomenon.

10.  In Figure 67, and 8, the authors need to specify the directional view (cross-sectional/surface or relevant) of the “before fracture” and “after fracture” SEM images. It is not obvious by looking at the images. Provide a schematic representation if that would help explaining the direction of the sample in which the imaging was conducted.

11.  Please use the comment in point 8 to address Figure 9. Lacking essential sample and statistical information.

12.  For the various spraying temperature and pressure conditions of polyurea, what was the constant pressure and constant temperature, respectively? Please include the information in the main manuscript.

13.  The authors claim that the best tensile characteristics can be obtained at 17.25 MPa and 70 deg. C. I am surprised to see that the authors haven’t prepared a sample under said conditions to demonstrate and endorse their claims. Were any such experiments conducted? What was the outcome? If not, the authors are expected to show the results for the same.

14.  The mathematical treatment of the fracture model appears abruptly towards the end of the manuscript. For a broad reader, the authors need to clearly outline the context for using this model and the understanding gained. Moreover, the variables used in various equations have not been explained, rendering it difficult to understand the model and its results. The authors are expected to clearly elucidate the various terms used and the treatments used for clarity of thoughts.

Round 2

Reviewer 1 Report

The authors have addressed my comments, and I believe the article can be accepted for publication in its present form.

Reviewer 2 Report

I thank the authors on their sincere effort to incorporate significant changes to the manuscript to improve its clarity and impact.